# META-CoT: GENERALIZABLE CHAIN-OF-THOUGHT PROMPTING IN MIXED-TASK SCENARIOS WITH LARGE LANGUAGE MODELS

## ABSTRACT

Large language models (LLMs) have unveiled remarkable reasoning capabilities by exploiting chain-of-thought (CoT) prompting, which generates intermediate reasoning chains to serve as the rationale for deriving the answer. However, current CoT methods either simply employ general prompts such as *Let's think step by step*, or heavily rely on handcrafted task-specific demonstrations to attain preferable performances, thereby engendering an inescapable gap between performance and generalization. To bridge this gap, we propose Meta-CoT, a generalizable CoT prompting method in mixed-task scenarios where the type of input questions is unknown. Meta-CoT firstly categorizes the scenario based on the input question and subsequently constructs diverse demonstrations from the corresponding data pool in an automatic pattern. Meta-CoT simultaneously enjoys remarkable performances on ten public benchmark reasoning tasks and superior generalization capabilities. Notably, Meta-CoT achieves the state-of-the-art result on SVAMP (93.7%) without any additional program-aided methods. Our further experiments on five out-of-distribution datasets verify the stability and generality of Meta-CoT. Code is available at `Anonymous`.

## 1 INTRODUCTION

Large language models (LLMs) (Brown et al., 2020; Scao et al., 2022; Thoppilan et al., 2022; Chowdhery et al., 2022; Touvron et al., 2023; OpenAI, 2023) have exhibited commendable capabilities on complex reasoning by virtue of chain-of-thought (CoT) prompting (Wei et al., 2023). CoT prompting entails the generation of intermediate reasoning chains that serve as the rationale before deriving the answer.

Current CoT prompting methods predominantly fall into two categories, which we dub as *General Zero-Shot-CoT* and *Specific Few-Shot-CoT*, respectively. The former leverages general prompts such as *Let's think step by step* and appends them directly to the input question, aiming to summon up the step-by-step reasoning potential from LLMs (Kojima et al., 2023; Yang et al., 2023). The latter provides task-specific input-output pairs as in-context demonstrations and puts them before the input question, for the purpose of instructing LLMs to carry out multi-step reasoning with elaborately selected demonstrations (Wei et al., 2023; Zhang et al., 2023; Wan et al., 2023; Diao et al., 2023).

Briefly, there are two major limitations in previous studies. On one hand, the *General Zero-Shot-CoT* pattern is endowed with favorable generalization ability as it does not need any task-related exemplars, but it often pales in terms of performance when compared with the few-shot pattern. On the other hand, the *Specific Few-Shot-CoT* pattern heavily leans on task-specific demonstrations to attain superior performances, yet fails to bear on decent generalization ability. Although recent works have made progress by either alleviating manual labor (Zhang et al., 2023) or promoting the quality of demonstrations (Arora et al., 2023; Wan et al., 2023; Diao et al., 2023), all of them rest on the task-associated perspective thus far.

Nevertheless, in practical applications, LLMs tend to confront situations of mixed types of questions, where it cannot be clearly identified which task the question belongs to. On these occasions, it is neither reasonable to improvise several task-related examples by hand nor possible to manually search for which task it refers to, not to mention that the question encountered in actual cases is not

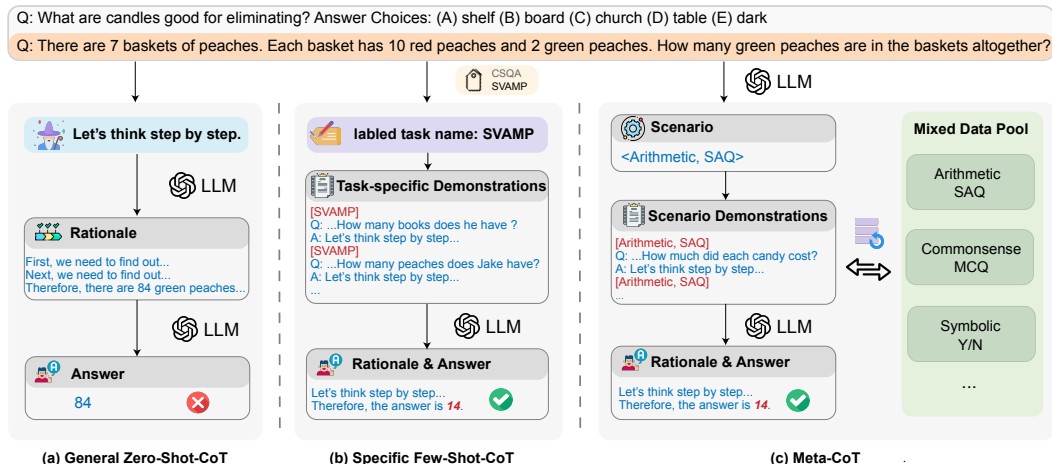

Figure 1: Comparison with existing paradigms of CoT prompting. General zero-shot-CoT and specific few-shot-CoT are from Kojima et al. (2023) and Wei et al. (2023), respectively.

even from a pre-defined collection of tasks. Besides, naive use of general trigger prompts is likely to result in performance degradation as the lack of templated rationales often leads to spurious reasoning steps (Wan et al., 2023). Therefore, there exists an inescapable gap between performance and generalization, especially in realistic mixed-task scenarios. To mitigate this gap, a potential strategy is to explore the trade-off area between generality and performance while ensuring certain practical applicability.

Motivated by the above ideas, we propose Meta-CoT: a generalizable CoT prompting method in mixed-task scenarios where the type of input questions is unknown. Meta-CoT comprises three phases: firstly, it gathers questions of various reasoning types from a collection of reasoning tasks and samples distinct questions as in-context learning (ICL) demonstrations. Those ICL demonstrations are used to categorize the scenario of the input question. Secondly, it automatically constructs diverse demonstrations from the corresponding data pool based on the classified scenario obtained in the first phase. Thirdly, it performs a final inference on the input question with the demonstrations elaborated in the second phase and delivers the feedback to the data pool.

We evaluate our proposed Meta-CoT on ten benchmark reasoning tasks including: (i) arithmetic reasoning (MultiArith (Roy & Roth, 2015), GSM8K (Cobbe et al., 2021), AddSub (Hosseini et al., 2014), AQUA-RAT (Ling et al., 2017), SingleEq (Koncel-Kedziorski et al., 2015), SVAMP (Patel et al., 2021)); (ii) commonsense reasoning (CSQA (Talmor et al., 2019), StrategyQA (Geva et al., 2021)); (iii) symbolic reasoning (Last Letter Concatenation, Coin Flip) (Wei et al., 2023). In addition, we further validate the stability and generalization of Meta-CoT on five out-of-distribution datasets including ARC-challenge (Clark et al., 2018), ASDiv (Miao et al., 2020), CSQA2.0 Talmor et al. (2021), Sports Understanding (Suzgun et al., 2022) and Creak (Onoe et al., 2021). Experimental results show that Meta-CoT simultaneously enjoys remarkable performances and superior generalization capabilities. Notably, Meta-CoT achieves the state-of-the-art result on SVAMP (**93.7%**) without any additional program-aided methods. Moreover, Meta-CoT achieves impressive performance on GSM8K (**89.92%**) even without in-context demonstrations from GSM8K itself.

To sum up, our work has three major contributions as follows:

(i) To the best of our knowledge, our work pioneers a novel setting of the mixed-task scenario for CoT prompting, which has significant practical application values.

(ii) We propose a generalizable CoT prompting method in mixed-task scenarios, which not only bridges the gap between performance and generalization but also unearths their in-between mutual synergy by gaining performance improvements in sync with achieving generality.

(iii) Our approach has shown impressive performance and superior generalization ability on a total of 15 in-distribution and out-of-distribution datasets. Notably, it achieves the state-of-the-art result on SVAMP (93.7%) without any additional program-aided methods.

Table 1: Typical CoT techniques (ICL: in-context learning; FT: fine-tuning; KD: knowledge distillation). Segment 1: fine-tuning techniques; Segment 2: in-context learning techniques. To the best of our knowledge, our work is the first to apply CoT prompting to mixed-task scenarios with enjoyable generality and superior performance without additional manual labor. In our work, we focus on in-context learning techniques, eliminating the burden of fine-tuning LLMs.

| Model | Training | Generality | w/o Manual Labor | w/ Input-related Info. |
|---|---|---|---|---|
| Fine-tune-CoT (Ho et al., 2022) | KD | ✗ | ✓ | ✗ |
| LoRAHub (Huang et al., 2023) | FT | ✓ | ✓ | ✗ |
| Zero-Shot-CoT (Kojima et al., 2023) | ICL | ✓ | ✓ | ✗ |
| Few-Shot-CoT (Wei et al., 2023) | ICL | ✗ | ✗ | ✓ |
| Self-Consistency-CoT (Wang et al., 2023) | ICL | ✗ | ✗ | ✓ |
| Least-to-Most Prompting (Zhou et al., 2023) | ICL | ✗ | ✗ | ✓ |
| Auto-CoT (Zhang et al., 2023) | ICL | ✗ | ✓ | ✓ |
| Active Prompt (Diao et al., 2023) | ICL | ✗ | ✗ | ✓ |
| OPRO (Yang et al., 2023) | ICL | ✗ | ✓ | ✗ |
| Meta-CoT (our work) | ICL | ✓ | ✓ | ✓ |

## 2 RELATED WORK

Two lines of research are key to our work: CoT prompting and cross-task generalization.

### 2.1 CHAIN-OF-THOUGHT PROMPTING

Recently, CoT prompting methods have pushed the multi-step reasoning abilities of LLMs to a remarkable aptitude by eliciting them to generate intermediate reasoning chains before deriving the final answer (Wei et al., 2023), of which some typical techniques are listed in Table 1. Currently, there are two flavors of research in CoT prompting: *General Zero-Shot-CoT* (Kojima et al., 2023) and *Specific Few-Shot-CoT* (Wei et al., 2023). The former merely appends a *general* prompt such as *Let's think step by step* to the input question, with the intuition that the step-by-step capabilities of LLMs can be conjured with simple natural language triggers. The latter leverages several task-*specific* input-output pairs as reasoning demonstrations and inserts them before the test question, in light of decent in-context learning capability of LLMs (Radford et al., 2019; Brown et al., 2020).

**General Zero-Shot-CoT.** LLMs have proven to be competent zero-shot reasoners by Kojima et al. (2023), which has greatly broadened the generalizability of CoT techniques and liberated the need to prepare task-specific examples in advance. While benefiting from its task-agnostic property, it often fails to excel at performance in comparison with its few-shot rivals (Wei et al., 2023; Zhang et al., 2023). In order to further boost the performance, recent works have laid emphasis on the optimization of triggering prompts (Yang et al., 2023). In their work, LLMs are employed as optimizers, and new prompts are progressively generated based on the past optimization history. Despite the augmented performance, the optimization process for prompts reverts to a task-specific problem, and for unseen test questions in real-world scenarios, it may not be advisable to use LLMs to optimize prompts on the fly.

**Specific Few-Shot-CoT.** Owing to the well-crafted in-context exemplars, Few-Shot-CoT achieves preferable performance, which consequently extends to a plethora of studies focusing on improvements upon it. According to the period of improvement, these studies are grouped into three categories: (i) pre-reasoning pattern; (ii) peri-reasoning pattern; and (iii) post-reasoning pattern.

For the pre-reasoning pattern, current research attends to either alleviating manual labor when selecting demonstrations (Zhang et al., 2023; Wan et al., 2023), or promoting demonstration quality (Creswell et al., 2023; Madaan & Yazdanbakhsh, 2022; Arora et al., 2023; Diao et al., 2023). Auto-CoT (Zhang et al., 2023) exploited the benefits of diversity in demonstrations and automatically constructed the demonstrations without the need for additional manual labor. Active-Prompt (Diao et al., 2023) underscored the significance of uncertainty by intentionally selecting the most uncertain questions for annotation and utilizing them as demonstrations. For the peri-reasoning pattern, recent studies concentrate on fine-grained reasoning processes such as problem decomposition (Zhou et al.,

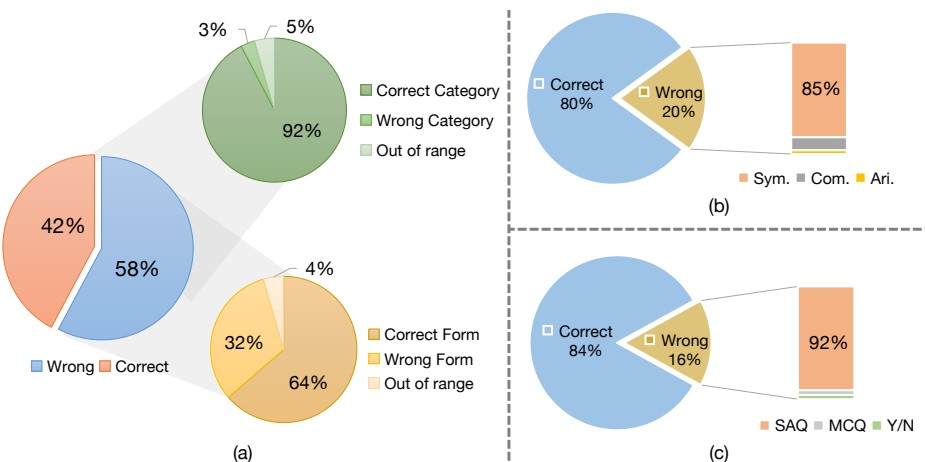

Figure 2: The ratio of wrong cases in task identification (a), ratio of wrong cases in category identification (b) and ratio of wrong cases falling into form identification (c).

2023; Press et al., 2022). Zhou et al. (2023) introduced least-to-most prompting, which reduced complex problems to sub-problems and then the sub-problems were solved sequentially. Self-ask (Press et al., 2022) specifically asked follow-up questions to the model and then answered them before responding to the initial question. For the post-reasoning pattern, related works principally enhanced the performance by verification (Weng et al., 2022; Lyu et al., 2023) or ensemble-like methods (Wang et al., 2023; Li et al., 2023; Wang et al., 2022; Yoran et al., 2023). Weng et al. (2022) computed an explainable answer verification score by taking turns masking the initial conditions and predicting their results. Wang et al. (2023) introduced a self-consistency decoding approach to sample multiple outputs of LLMs and then voted over the final answers.

However, the aforementioned works, which mainly hinge on task-associated exemplars, fail to step outside the task-specific framework to pursue generalizability. In turn, there is an upper bound to the performance that a general Zero-Shot-CoT method can achieve, thus leading the current CoT prompting to a dilemma. Our work, in contrast, manages to find a way out of this dilemma by intuitively carrying out an upstream scenario identification task, making our proposed Meta-CoT applicable in realistic mixed-task scenarios.

## 2.2 Cross-task Generalization

Cross-task generalization has been a long-standing research goal in natural language processing (NLP). The conventional pre-training and fine-tuning paradigm gains a foothold by pre-training on a large corpus of text to capture general knowledge and fine-tuning on specific tasks to acquire specific knowledge. Beyond this primitive paradigm, post pre-training and multi-task learning encourage further advancements in this research area. For instance, Yu et al. (2022) made progress in the science domain while Zhang & Zhao (2021) promoted the model's performance on dialogue-related tasks by introducing two novel training objectives to incorporate the dialogue-like features. Furthermore, typical multi-task learning frameworks promote models to learn shared representations across tasks to achieve task generalization. For example, MT-DNN (Liu et al., 2019) leveraged a few task-aware output modules to tailor the shared representations to each task. Notably, Zhang et al. (2022) proposed a task prefix guided multi-task pre-trainig framework, under the motivation that there are potential relationships among tasks which can be helpful for task generalization. Our work, consequently, is inspired by the discovery that data from different tasks may have similarities, thus sensibly partitioning mixed questions is likely to detect the mutual synergy between generalization and performance. More recent works such as ExT5 (Aribandi et al., 2022), T0 (Sanh et al., 2022) and FLAN (Wei et al., 2022) strived to convert a variety of tasks into an identical text-to-text format, so that models can be trained on those tasks jointly. LoraHub (Huang et al., 2023) leveraged the composability of LoRA (Low-Rank Adaption of LLMs) modules to promote the task generalization ability of LLMs. Our work, however, manages to effectuate task generalization through timely and user-friendly in-context learning without any training.

## 3 CHALLENGES OF GENERALIZABLE CoT IN MIXED-TASK SCENARIOS

Existing studies (Wei et al., 2023) commonly assume that the type of questions fed to the model is known and conduct each set of evaluations on the questions from the same dataset. However, a more realistic setting lies in mixed-task scenarios where the type of input questions is unknown and they come in an arbitrary manner. To address the mixed-task scenarios, we put forward a salient procedure, namely *scenario identification* to explore practical and efficient solutions in a plug-and-play fashion. Beforehand, we need to address the following two challenges: (i) How to effectively partition the mixed questions so that we can invoke the pre-defined solutions (e.g., scenario-wise ICL)? (ii) What information do LLMs need to know for efficient scenario identification?

### 3.1 PARTITIONING MIXED QUESTIONS

In the first place, we investigate how to effectively partition the mixed questions. Following Kojima et al. (2023); Zhang et al. (2023), we adopt questions from ten reasoning tasks. Those questions cover three categories including arithmetic, commonsense and symbolic reasoning and involve three forms encompassing short-answer, multiple-choice, and yes-or-no questions [1]. At the very beginning, we make a simple and naive attempt to test how well LLMs can identify various tasks. We randomly sample one question from each of the ten tasks. For each question, we retain the task name from which it originates so that we obtain ten *question-task* pairs, which we employ as in-context learning demonstrations for question type identification.

As can be seen from Figure 2, the identification accuracy is only 42%. We then analyze the wrong examples and find that 92% and 64% of them belong to the same category and form as the correct task respectively. The results demonstrate that LLMs are not qualified for distinguishing task names, but possess a high probability of correctly discriminating their categories or forms. We speculate that the underlying reason can be two-fold: on one hand, task names themselves are too abstract for LLMs to well perceive their differences through in-context learning alone. On the other hand, there exist potential similarities and correlations among tasks themselves (Zhang et al., 2022), which enlightens us to disclose more rational partitioning strategies.

Since the majority of cases that misidentify task names fall into the same category or form, we compare the identification accuracy with the following three variants of partitioning schemes: (i) Category-based scheme which separates mixed questions into diverse categories; (ii) Form-based scheme which segments data into different answer forms; (iii) <Category, Form>-based scheme which concurrently takes the two aspects into account. As is shown in the right parts of Figure 2, we find that for category- and form-based schemes, a particular group tends to dominate the wrong cases. For instance, 85% of wrong cases in category identification belong to the symbolic group. We discover that this is because the sampled *symbolic* group demonstrations do not cover *symbolic yes-or-no* question, thus hindering

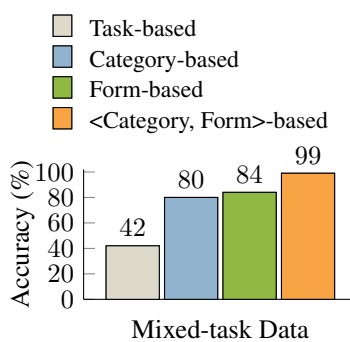

Figure 3: Identification accuracy (%) with different partitioning schemes.

LLMs from accurately identifying this missing type. As such, partitioning mixed questions based on both its **category and form** is a sensible strategy, which adequately considers the two major natures of question data. The results in Figure 3 show that this strategy reaches high accuracy (99%).

### 3.2 IDENTIFYING SCENARIOS

In this part, we analyze what information LLMs require for efficient scenario identification. We extract the questions (Q) from the original data files and obtain the corresponding rationales (CoT) and predicted answers (A) from the Zero-Shot-CoT log files from Kojima et al. (2023). Abiding by the <Category, Form>-based partitioning strategy discussed in Section 3.1, we consider four alternatives of input formats fed to LLMs for scenario identification: (i) [Q] which takes purely the question as input ; (ii) [Q, A] which concatenates the question and the corresponding predicted

---

[1]More data information is shown in Appendix A.1

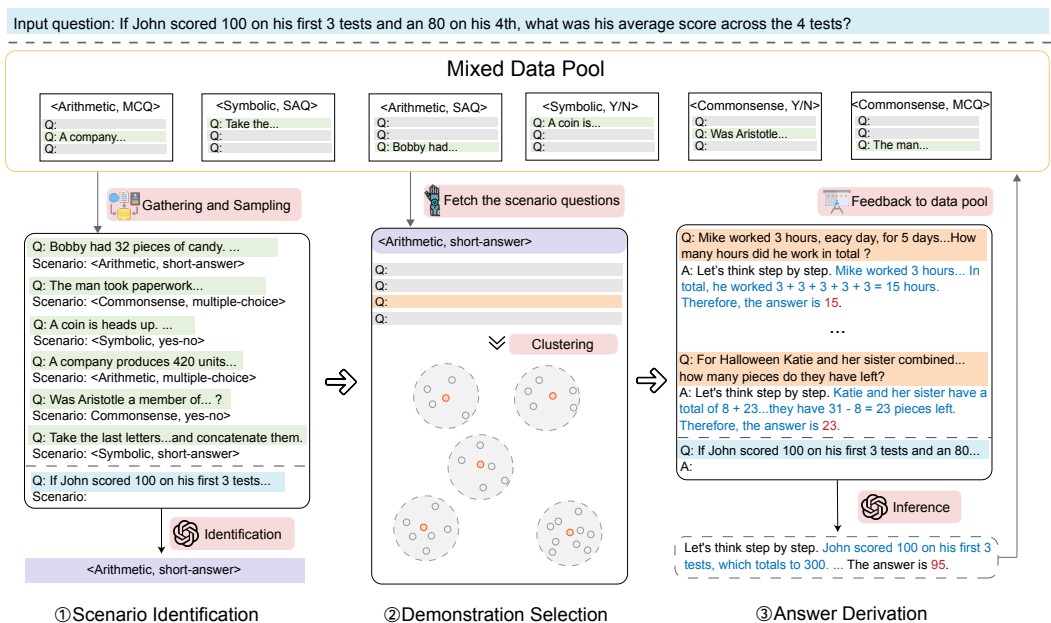

Figure 4: Overview of Meta-CoT, which consists of three phases: (i) *scenario identification*: categorizes the scenario of the input question (left); (ii) *demonstration selection*: fetches the ICL demonstrations for the categorized scenario (middle); (iii) *answer derivation*: performs the answer inference by feeding the LLM with the prompt comprising the fetched ICL demonstrations and the input question (right).

answer; (iii) [Q, CoT] which joins the question and the rationale together ; (iv) [Q, CoT, A] which sequentially combines the question, rationale and answer.

Results in Table 2 suggest that the question itself is sufficient for LLMs to perceive the scenario. Notably, the participation of CoT degrades the identification performance, which may reveal that LLMs only need to focus on the question itself and the rationales would distract LLMs, thus leading to identification errors. Therefore, the **question-only pattern [Q]** is a satisfactory input option for scenario identification with decent accuracy and generality.

Table 2: Identification accuracy (%) with different input formats.

| Input format | Generality | Accuracy |
|---|---|---|
| [Q] | ✓ | **99.00** |
| [Q, A] | ✗ | 96.40 |
| [Q, CoT] | ✗ | 90.30 |
| [Q, CoT, A] | ✗ | 91.10 |

## 4 META-COT

This section introduces Meta-CoT, which is illustrated in Figure 4. On a high level, Meta-CoT consists of three phases: (i) *scenario identification*: categorizes the scenario of the input question; (ii) *demonstration selection*: fetches the ICL demonstrations for the categorized scenario; (iii) *answer derivation*: performs the answer inference by feeding the LLM with the prompt comprising the fetched ICL demonstrations and the input question. We detail these phrases as follows.

### 4.1 SCENARIO IDENTIFICATION

Given an input question $q_{in}$, the goal of the scenario identification phase is to categorize the scenario, e.g., the type of the question. To this end, we first prepare a few ICL demonstrations, each of which consists of a question $q_i$ and its scenario $s_i$. The ICL demonstrations will be concatenated with $q_{in}$ to prompt the LLM to infer the question scenario. At the very beginning, we leverage public off-the-shelf datasets and obtain $n$ data groups based on the <category, form >partitioning strategy to construct the ICL demonstrations. Now that we have $n$ data groups $[D_1, D_2, \ldots, D_n]$ as a mixed questions pool *MP*, we randomly sample one question from each data group and obtain a set of

questions $[q_1, q_2, \ldots, q_n]$, with $q_i \in D_i$. Let $s_i$ represent the scenario name for the data group $D_i$. The demonstration $d_i$ for data group $D_i$ is formed by: $d_i = [\text{Q: } q_i, \text{Scenario: } s_i]$. We run such a process for each data group to have $n$-shot demonstrations: $P_{icl} = [d_1, d_2, \ldots, d_n]$. Similarly, the prompted input for identification $P_{ide}$ can be formulated as $[\text{Q: } q_{in}, \text{Scenario: }]$. Finally, we concatenate the demonstrations and the prompted input together: $[P_{icl}, P_{ide}]$ and feed it into LLMs to predict the scenario $s_{in}$ for $q_{in}$.

## 4.2 DEMONSTRATION SELECTION

After categorizing the scenario $s_{in}$ for the input question $q_{in}$, we are able to construct scenario-wise demonstrations for in-context learning. Given the scenario $s_{in}$ for the input question obtained in Section 4.1, we fetch the corresponding scenario data group $D_{in} \in [D_1, D_2, \ldots, D_n]$. Therefore, we have the questions in $D_{in}$ under the same scenario with $q_{in}$. Then, we construct the few-shot demonstrations by sampling a few representative questions by $k$-means clustering and invoking Zero-Shot-CoT to obtain the reasoning chains following Auto-CoT (Zhang et al., 2023).

Concretely, we leverage Sentence-BERT (Reimers & Gurevych, 2019) to obtain a vector representation for each candidate question in $D_{in}$. Afterward, we perform $k$-means clustering over the acquired contextualized representations. For each cluster $i$, we sort the questions in ascending order by distance from the cluster center. Then we iterate over the sorted question list and apply Zero-Shot-CoT to the current question, namely adding *Let's think step by step* after the question, to get the rationale and predicted answer. Next, we follow prior works (Wei et al., 2023; Zhang et al., 2023) and conduct simple filtering operations [2] on the question and rationale, which help obtain more effective demonstrations. Once the *question-rationale* pair is retained under the filtering operation, we stop functioning on other questions in cluster $i$. As a result, we manage to collect a total of $k$ representative and high-quality demonstrations for $D_{in}$: $[(q_{re}^1, r_{re}^1, a_{re}^1), (q_{re}^2, r_{re}^2, a_{re}^2), \ldots, (q_{re}^k, r_{re}^k, a_{re}^k)]$, where $r_{re}^i$ and $a_{re}^i$ refer to the rationale and predicted answer of $q_{re}^i$ by invoking Zero-Shot-CoT.

## 4.3 ANSWER DERIVATION

Now that we have $k$ typical demonstrations of the formerly classified scenario $s_{in}$, we execute a final inference to obtain the answer to $q_{in}$. Concretely, we construct each demonstration $d_{re}^i$ by: $d_{re}^i = [\text{Q: } q_{re}^i, \text{A: } r_{re}^i, a_{re}^i]$ where $q_{re}^i$, $r_{re}^i$, and $a_{re}^i$ are from $D_{in}$. Then we prepare the templated input prompt for inference by $P_{inf} = [\text{Q: } q_{in}, \text{A: } \texttt{Prompt}]$, where $\texttt{Prompt}$ refers to simple triggers such as *Let's think step by step*. After that, the concatenated demonstrations $[d_{re}^1, d_{re}^2, \ldots, d_{re}^k]$ are inserted before the input prompt $P_{inf}$, which is eventually delivered to LLMs to derive the rationale $r_{in}$ and answer $a_{in}$ of input question $q_{in}$. Meanwhile, we obtain a new triple of the input question, rationale and answer $(q_{in}, r_{in}, a_{in})$, which is sent back to the identified data group $D_{in}$ to update the mixed questions pool *MP*.

## 5 EXPERIMENTS

This section will describe our experimental setup and present the main results.

## 5.1 SETUP

**Tasks and Datasets.** Our method is evaluated on 10 in-distribution benchmark datasets and 5 out-of-distribution datasets. The in-distribution datasets are from three categories of reasoning tasks: (i) arithmetic reasoning (MultiArith (Roy & Roth, 2015), GSM8K (Cobbe et al., 2021), AddSub (Hosseini et al., 2014), AQUA-RAT (Ling et al., 2017), SingleEq (Koncel-Kedziorski et al., 2015), SVAMP (Patel et al., 2021)); (ii) commonsense reasoning (CSQA (Talmor et al., 2019), StrategyQA (Geva et al., 2021)); (iii) symbolic reasoning (Last Letter Concatenation, Coin Flip) (Wei et al., 2023). The five out-of-distribution datasets include: ARC-challenge (Clark et al., 2018), ASDiv (Miao et al., 2020), CSQA2.0 Talmor et al. (2021), Sports Understanding (Suzgun et al., 2022) and Creak (Onoe et al., 2021).

---

[2]More details are attached in Appendix B.1

Table 3: Accuracy (%) on ten in-distribution reasoning datasets. Segment 1: ICL methods without CoT; Segment 2: Task-specific CoT approaches; Segment 3: CoT techniques with generalization. † indicates the experiment is based on GPT-4, otherwise GPT-3.5-Turbo is employed by default. Results in **bold** and underline are the best and second-best performances respectively.

| Method | AQuA | MultiArith | AddSub | GSM8K | SingleEq | SVAMP | Letter | Coin | Strategy | CSQA | Avg. |
|---|---|---|---|---|---|---|---|---|---|---|---|
| Zero-Shot | 29.1 | 67.2 | 84.5 | 15.9 | 83.1 | 67.9 | 4.8 | 44.0 | 65.3 | 74.3 | 53.6 |
| Few-Shot | 33.1 | 87.5 | 86.6 | 22.8 | 89.0 | 79.1 | 7.2 | 64.4 | 62.3 | 81.0 | 61.3 |
| Few-Shot-CoT | 54.3 | 97.3 | 89.1 | 73.8 | 92.9 | 81.9 | 73.2 | 99.0 | 63.7 | 78.0 | 80.3 |
| Auto-CoT | 49.6 | 99.3 | 89.6 | 75.9 | 92.3 | 84.6 | 81.2 | **100.0** | 64.6 | 72.2 | 80.9 |
| Zero-Shot-CoT | 51.6 | 94.7 | 84.2 | 71.2 | 91.1 | 78.4 | 85.8 | 99.0 | 62.6 | 69.9 | 78.8 |
| General-CoT | 46.9 | 98.7 | 87.9 | 74.1 | 92.9 | 83.8 | 75.2 | **100.0** | 63.4 | 72.2 | 79.5 |
| Meta-CoT | 54.7 | **99.7** | 90.9 | 72.6 | **93.5** | 88.6 | 77.2 | **100.0** | 64.5 | 72.4 | 81.4 |
| Meta-CoT† | **72.8** | 99.0 | **91.9** | **89.9** | 92.3 | **93.7** | **90.2** | **100.0** | **74.1** | **86.4** | **89.0** |

**Implementation.** We utilize the popular and publicly available GPT-3.5-Turbo and GPT-4 (OpenAI, 2023) from OpenAI API [3]. Experimental results are based on GPT-3.5-Turbo by default unless otherwise specifically marked. The original mixed questions pool *MP* is constructed based on the 10 in-distribution datasets. The number of data groups $n$ is 6 according to the partitioning scheme discussed in Section 3.1. Following Wei et al. (2023), the number of demonstrations $k$ is 8 except for <arithmetic, multiple-choice questions> and <symbolic, short-answer questions > (4), <commonsense, multiple-choice questions> (7) and <commonsense, yes-or-no questions> (6).

**Baselines.** We compare Meta-CoT with 6 baselines, which can be divided into three groups: (i) ICL methods without CoT prompting including Zero-Shot (Kojima et al., 2023) and Few-Shot (Brown et al., 2020); (ii) task-specific CoT approaches involving Few-Shot-CoT (Wei et al., 2023) and Auto-CoT (Zhang et al., 2023); (iii) CoT techniques with generalization referring to Zero-Shot-CoT (Kojima et al., 2023) and General-CoT. General-CoT is a strong baseline that we specifically devise for generalization comparison. It randomly collects one demonstration from each partitioned question group in our mixed data pool (*MP*) and then leverages the gathered demonstrations as a generic inference prompt for all the input data. [4]

## 5.2 MAIN RESULTS

**Performance of Meta-CoT on 10 in-distribution datasets** Table 3 presents the results on ten in-distribution reasoning tasks. Notably, Meta-CoT achieves a state-of-the-art result on SVAMP (93.7%) without any additional program-aided methods. Meta-CoT also attains impressive performance on GSM8K without in-context demonstrations from GSM8K itself. Furthermore, Meta-CoT towers above all the baseline methods from different angles. On one hand, compared with two typical task-specific CoT approaches, Meta-CoT not only surpasses them in performance but also enjoys the generalizable property, which means that the input question with an unknown type can be adapted to our method in an automatic and labor-free pattern. On the other hand, while the general CoT techniques both witness performance degradation (i.e., 80.9%→78.8/79.5%), Meta-CoT stands out by continually boosting the performance (i.e., 80.9%→81.4%), thus shedding light on the mutual synergy between performance and generalization of LLMs.

**Performance of Meta-CoT on five out-of-distribution datasets** As our work aims to accomplish a generalizable CoT prompting method in mixed-task scenarios, we further conduct experiments on 5 out-of-distribution datasets to verify its generality. We observe from Table 4 that our approach is capable of achieving a decent performance while maintaining favorable stability. The results certify the applicability of Meta-CoT to realistic situations where the incoming data is not defined by a certain type. Besides, we surprisingly discover that comparable results are yielded with the demonstrations of <*commonsense, yes-or-no questions* > scenario. We analyze that it is probably due to the broad coverage of commonsense knowledge that assists in the generality of LLMs.

---

[3] https://openai.com/blog/openai-api
[4] More details are presented in Appendix B.2.

Table 4: Accuracy (%) on five out-of-distribution datasets. SAQ: short-answer question; MCQ: multiple-choice question; Y/N: yes-or-no question. We report the mean (Avg.) and standard deviations (Std.). We calculate Std. based on different question groups. Segment 1: Methods that leverage demonstrations of a specified scenario; Segment 2: Our Meta-CoT method. Results in **bold** and underline are the best and second-best performances respectively.

| Method | Creak | Sports | CSQA2.0 | ASDiv | ARC-c | Avg.± Std. |
|---|---|---|---|---|---|---|
| Symbolic, SAQ | 10.8 | 58.5 | 22.4 | 73.2 | 66.6 | 56.8±22.9 |
| Symbolic, Y/N | 28.3 | 22.6 | 33.3 | 73.3 | 60.9 | 54.1±23.4 |
| Arithmetic, SAQ | 8.6 | 43.6 | 16.7 | 77.2 | 67.6 | 55.9±28.9 |
| Arithmetic, MCQ | 18.8 | 59.1 | 28.5 | **77.3** | 70.0 | 61.2±22.5 |
| Commonsense, Y/N | **85.7** | **83.1** | **65.2** | 71.7 | 76.6 | 75.4 ± 3.3 |
| Commonsense, MCQ | 22.5 | 25.5 | 23.5 | 74.0 | **77.9** | 58.6±30.2 |
| Meta-CoT | 85.1 | **83.1** | 62.3 | 77.1 | 77.6 | **77.2±0.4** |

# 6 ANALYSIS

## 6.1 METHODS OF CONSTRUCTING CoT DEMONSTRATIONS

Since our work is situated in realistic mixed-task scenarios, accessing high-quality demonstrations in a labor-saving pattern is of crucial importance. Accordingly, we select two representative labor-free sampling methods for comparison: (i) Similarity-based which retrieves the most top-$k$ similar questions based on cosine similarity; (ii) Randomness-based which randomly samples $k$ demonstrations for each input question. Results in Table 5 show our proposed Meta-CoT performs best, illustrating the importance of diversity in demonstrations.

Table 5: Accuracy (%) of different demonstration construction methods.

| Method | AQuA | Strategy | Coin |
|---|---|---|---|
| Meta-CoT | **54.7** | **64.5** | **100.0** |
| w/ similarity | 49.6 | 64.1 | 99.2 |
| w/ randomness | 52.0 | 61.2 | 99.0 |

## 6.2 EFFECT OF SCENARIO IDENTIFICATION

In order to further explore the effect of scenario identification which plays a key role in generalization, we discard this identification phase and adopt an idealized strategy in which we assume that the model is given the gold scenario. Results in Table 6 reveal that only a trivial improvement is detected even with the correct scenario given (70.2% $\rightarrow$ 70.6%). This indicates that our method potentially arouses the self-determination ability of LLMs without the need for manual intervention.

Table 6: Effect of scenario identification. We study the cases where the correct scenario for the input question is given and them compare them with our method, which adaptively predicts the scenario.

| Method | AQuA | Strategy | ASDiv | Creak | CSQA2.0 | ARC-c | Avg. |
|---|---|---|---|---|---|---|---|
| Meta-CoT | 54.7 | 64.5 | 77.1 | 85.1 | 62.3 | 77.6 | 70.2 |
| w/ correct scenario | 52.8 | 65.0 | 77.2 | 85.7 | 65.2 | 77.9 | 70.6 |

# 7 CONCLUSION

In this work, we initially put forward a novel setting with significant application values, namely mixed-task scenarios where the type of input question is unknown. Upon this challenging setting, we propose Meta-CoT, a generalizable CoT prompting mechanism that first performs scenario identification based on the input data and then automatically constructs corresponding demonstrations for ICL. Evaluation results on a total of 15 in-distribution and out-of-distribution datasets demonstrate the impressive performance and superior generalization ability of our proposed approach. While most existing works focus on either promoting performance or pursuing generality, we open up a pioneering perspective to bridge the two aspects in a simple and practical manner.

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

## A    DATASET INFORMATION

### A.1    IN-DISTRIBUTION DATASETS

Our method is evaluated on 10 in-distribution benchmark datasets that cover three categories including arithmetic, commonsense and symbolic tasks and involve three forms encompassing short-answer, multiple-choice, and yes-or-no questions. The corresponding categories and forms of these datasets are shown in Table 7.

• **Arithmetic Reasoning**: we choose the following six datasets: (i) MultiArith (Roy & Roth, 2015), (ii) GSM8K (Cobbe et al., 2021), (iii) AddSub (Hosseini et al., 2014), (iv) AQUA-RAT (Ling et al., 2017), (v) SingleEq (Koncel-Kedziorski et al., 2015), and (vi) SVAMP (Patel et al., 2021). MultiArith, AddSub, and SingleEq come from the Math World Problem Repository (Koncel-Kedziorski et al., 2016), while the other three are from more contemporary benchmarks. Among them, all the arithmetic datasets belong to short-answer form except for AQUA-RAT which is in multiple-choice format.

• **Commonsense Reasoning**: we take the following two datasets into account: (i) CSQA (Talmor et al., 2019) and StrategyQA (Geva et al., 2021). CSQA poses questions with complicated semantics that frequently necessitate prior knowledge reasoning (Talmor et al., 2019). StrategyQA requires models to conduct multi-hop reasoning during inference (Geva et al., 2021). CSQA is in multiple-choice form whereas StrategyQA belongs to the yes-or-no format.

• **Symbolic Reasoning**: we employ the typical datasets Last Letter Concatenation and Coin Flip from Wei et al. (2023), which are in short-answer and yes-or-no form respectively. Last Letter Concatenation asks the model to concatenate the last letters of each word. Coin Filp requires the model to answer whether a coin heads up after a series of actions of either flipping or not flipping the coin.

Table 7: Information of 10 in-distribution datasets (Ari.: arithmetic; Com.: commonsense and Sym.: symbolic; SAQ: short-answer question; MCQ: multiple-choice question; Y/N: yes-or-no question).

| Task | MultiArith | GSM8K | AddSub | AQuA | SingleEq | SVAMP | CSQA | Strategy | Letter | Coin |
|------|-----------|-------|--------|------|----------|-------|------|----------|--------|------|
| Category | Ari. | Ari. | Ari. | Ari. | Ari. | Ari. | Com. | Com. | Sym. | Sym. |
| Form | SAQ | SAQ | SAQ | MCQ | SAQ | SAQ | MCQ | Y/N | SAQ | Y/N |
| Size | 600 | 1319 | 395 | 254 | 508 | 1000 | 1221 | 2290 | 500 | 500 |

### A.2    OUT-OF-DISTRIBUTION DATASETS

Table 8 presents the information of 5 out-of-distribution datasets we use to evaluate our method, including ARC-challenge (Clark et al., 2018), ASDiv (Miao et al., 2020), CSQA2.0 Talmor et al. (2021), Sports Understanding (Suzgun et al., 2022) and Creak (Onoe et al., 2021). ARC-challenge (Clark et al., 2018) requires the model to cope with English language exam questions that span several grade levels as indicated in the files, which is in short-answer form. ASDiv (Miao et al., 2020) evaluates the model's capability of answering English math word problems in various language patterns and problem types. CSQA2.0 (Talmor et al., 2021) and Creak (Onoe et al., 2021) ask the model to judge the assertions about everyday commonsense knowledge as correct or incorrect. Sports Understanding (Suzgun et al., 2022) demands the model to determine whether a statement relating to sports is plausible or not.

Table 8: Information of 5 out-of-distribution datasets (SAQ: short-answer question; MCQ: multiple-choice question; Y/N: yes-or-no question).

| Task | ASDiv | Creak | Sports | CSQA2.0 | ARC-c |
|------|-------|-------|--------|---------|-------|
| Category | Arimetic | Commonsense | Commonsense | Commonsense | Commonsense |
| Form | SAQ | Y/N | Y/N | Y/N | SAQ |
| Size | 2305 | 1371 | 1000 | 2541 | 299 |

# B  EXPERIMENTAL DETAILS

## B.1  IMPLEMENTATION DETAILS

**Filtering operations in Demonstration Selection.**  We follow the works from (Wei et al., 2023; Zhang et al., 2023) and leverage simple criteria to filter the *question-rationale* pair as follows: the question needs to be no more than 60 tokens and the rationale should not exceed 5 reasoning steps. The reasoning step is identified by counting the number of \n tokens in the rationales. The \n token is often employed by Zero-Shot-CoT for separating reasoning steps. The objective of this filtering strategy is to seek simple heuristics by sampling simpler questions and rationales.

## B.2  BASELINE METHODS

We introduce the baselines methods in detail.

- **ICL methods without CoT**: Zero-Shot (Kojima et al., 2023) concatenates a test question with the prompt "A:" as the LLM input. Few-Shot (Brown et al., 2020) has the same LLM input as Zero-Shot except for several additional templated demonstrations as: [Q: q, A: The answer is a] before the test question, where q and a are manually crafted questions and answers.

- **Task-specific CoT approaches.**: Few-Shot-CoT (Wei et al., 2023) follows similar patterns as Few-Shot but differs in that rationales are inserted before deriving the answer. Auto-CoT (Zhang et al., 2023) divides questions of a given dataset into a few clusters, samples a representative question from each cluster, and constructs its reasoning chain using Zero-Shot-CoT with simple heuristics.

- **CoT techniques with generalization**: Zero-Shot-CoT (Kojima et al., 2023) simply inserts the prompt *Let's think step by step* after a test question to conduct inference, which rids the necessity of handcrafted task-wise demonstrations. We also compare our method with a strong baseline General-CoT, in which the in-context demonstrations for inference come from distinct question groups.

# C  FURTHER ANALYSIS

Table 5 demonstrates the complete results of different demonstration sampling methods on 6 well-partitioned data groups. We find that the diversity-based strategy performs best overall.

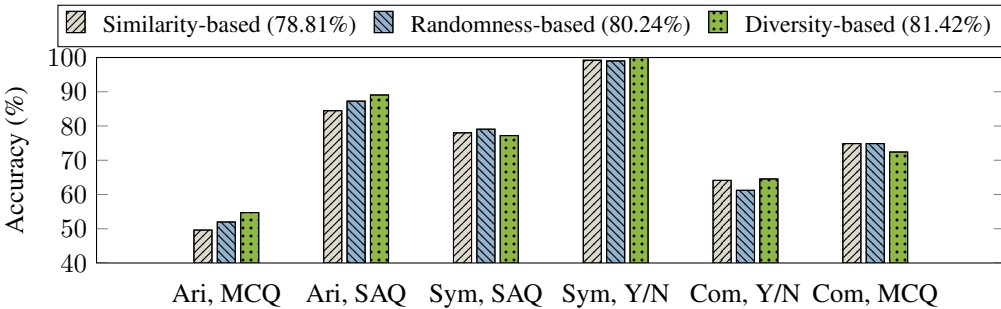

Figure 5: Results of different demonstration sampling methods on 6 data groups (Ari: arithmetic; Com: commonsense and Sym: symbolic).

