# OpenReview forum: "Meta-CoT: Generalizable Chain-of-Thought Prompting in Mixed-task Scenarios with Large Language Models"
_ICLR.cc/2024/Conference — Submitted to ICLR 2024_

### Official Review · Reviewer_JvEc · 2023-10-28

**Soundness:** 2 fair
**Presentation:** 3 good
**Contribution:** 1 poor
**Rating:** 3
**Confidence:** 5

**Summary:**

This paper presents a new prompting technique, dubbed Meta-CoT, that aims at combining the generalisability of zero-shot CoT and the performance benefits of few-shot in-context learning. It also proposes mixed task, a new problem setting in which a number of different problem categories are given in arbitrary order.

Meta-CoT firstly detects the category of a given question, and then selects exemplars from the predicted category, and finally prompts the LLM to obtain the final answer.

Experiments are performed on a collection of 10 in-domain datasets as well as 5 out-of-domain datasets, in which Meta-CoT is compared with a number of baselines.

**Strengths:**

* CoT prompting is an active area of research and important to elicit performance from LLMs.

**Weaknesses:**

* The proposed method is too simple and very specifically tied to the proposed mixed-task scenario.

* The performance of the proposed method is not strong. It performs on par with existing, and simpler methods when applied to the same LLM (GPT 3.5). The strong performance is achieved on GPT-4, thus, cannot be attributed to Meta-CoT.

**Questions:**

* I find the proposed mixed-task scenario not convincing as a challenging problem. The very simple technique proposed in Sec. 3.1 achieves 99% accuracy. Given this result, the mixed-task scenario essentially degenerates to the single-task scenario, isn't it?

* At the end of Sec. 4.3, what is the purpose of updating the mixed question pool $MP$?

* The performance on out-of-distribution experiments (Table 4) is not compared with any other baseline. This doesn't give evidence to the performance level of Meta-CoT.

---

### Official Review · Reviewer_FQzo · 2023-10-31

**Soundness:** 1 poor
**Presentation:** 1 poor
**Contribution:** 2 fair
**Rating:** 3
**Confidence:** 4

**Summary:**

This paper studies combining chain-of-thought (CoT) demonstrations from multiple datasets to construct an in-context (few-shot) prompt with CoT exemplars. First, a few-shot prompt is used to identify a "scenario". Then, the closest questions by Sentence-BERT are selected, and zero-shot demonstrations are generated for them (with "let's think step by step" prompts). Finally, the questions and generated CoT demonstrations are concatenated to create a few-shot prompt.

Experiments are performed on 10 in-distribution datasets and 5 out-of-distribution datasets.

**Strengths:**

1. Extensive experiments on multiple datasets
2. High quantitive results on in-domain tasks
3. No need for human-written CoT demonstrations

**Weaknesses:**

1. OOD results are not as high, questioning the claimed generalizability
2. The zero-shot generated CoT and answers for the demonstrations might have mistakes
3. More baselines and ablations are needed. Why are the methods from Table 1 not evaluated?
4. The main results table (table 3) mixes GPT-4 and GPT-3.5-Turbo in a confusing way since only the proposed method is using GPT-4.
5. The paper is hard to follow and full of hand wavy claims, unclear descriptions of methodology, and imprecise terminology. For example, what exactly is "scenario"? Why is the method called "meta"? How exactly are the results for the analysis in Figures 2 and 4 obtained?

**Questions:**

1. Are the gold answers of the questions used for demonstrations used in any way? Have you try comparing k-shot with gold answer with the CoT results?
2. Table 6 might indicate that the "scenario" identification is not important at all. Have you tried skipping the scenario identification step?
3. What sampling configuration was used? (temperature etc.)

---

### Official Review · Reviewer_uTXx · 2023-11-02

**Soundness:** 2 fair
**Presentation:** 2 fair
**Contribution:** 2 fair
**Rating:** 3
**Confidence:** 4

**Summary:**

The paper introduces Meta-Cot, a dynamic few-shot example selection strategy aiming to overcome the limitations of existing CoT strategies that rely on fixed few-shot prompts or generic instructions. This addresses a key limitation of existing few-shot prompting approaches: the nature of the task is not known in advance in the real world, making few-shot prompting approaches suboptimal. Meta-Cot involves two steps: i) given the test question, identify one out of k hardcoded scenarios supported by the system (e.g., that the question is symbolic and requires a binary answer), and ii) select in-context examples for the scenario. The fetched scenario is used to run inference with the LLM. Experiments on several benchmark datasets show promising results.

**Strengths:**

The paper tackles an important problem and highlights some crucial shortcomings of current prompting methods.

**Weaknesses:**

- The central claim of generalizing to new questions with a generic prompt is not fully substantiated, as the system only supports 6 hardcoded categories and shows suboptimal performance in out-of-domain scenarios. Additionally, the datasets labeled as out-of-domain (OOD) are not genuinely OOD. For instance, while CSQA and GSM are considered in-domain, CSQA v2.0 and ASDiv are deemed OOD (it is worth noting that using GSM prompts for ASDiv is a common practice). Finally, the OOD performance is suboptimal, with only ASDIV (1 out of 5) datasets showing any gains.

- Regarding the experimental results, Table 3 presents GPT-3 and GPT-4 numbers from MetaCot (the proposed method) together, which could be misleading. Comparison with the gpt-3.5-turbo-based method (AutoCot) shows that the gap is 0.5 points. Comparing this with the data requirements that the proposed method imposes brings the approach's utility into question. Overall, the experimental results are weak and do not justify the rather complex setup used for sample selection.

- The paper reports 89% with GSM-8k. However, this number is with GPT-4, which has been fine-tuned on GSM-8k. Further, the number of GSM-8k without any finetuning is 92% ([gpt-4 technical report](https://arxiv.org/pdf/2303.08774.pdf)).

**Questions:**

- Why doesn't the comparison in Table 4 include specific few-shot prompts and 0-shot CoT? I believe that will be the real test of the Meta-CoT's efficacy in OOD scenarios?


- (Minor) Missing citation for dynamic example selection:
Liu, Jiachang, Dinghan Shen, Yizhe Zhang, Bill Dolan, Lawrence Carin, and Weizhu Chen. "What Makes Good In-Context Examples for GPT-$3 $?." arXiv preprint arXiv:2101.06804 (2021).

---

### Meta-Review · Area_Chair_Zi3J · 2023-12-10

**Metareview:**

The authors propose the idea of automatically selecting good CoT examples by identifying the category associated with the input query. They coin this approach meta-CoT (akin to meta-learning that solves multiple tasks). While the idea is certainly promising, reviewers found multiple major issues with the paper such as approach supporting 6 query categories, concerns with out-of-distribution problems, whether mixed-task is inherently challenging etc. These highlight that the paper is not ready for publication at this point. I recommend authors to improve their manuscript based on reviewer suggestions.

**Justification For Why Not Higher Score:**

N/A

**Justification For Why Not Lower Score:**

N/A

---

### Decision · Program_Chairs · 2024-01-16

Reject